# Faith-Based Spiritual Intervention for Persons with Depression: Preliminary Evidence from a Pilot Study

**DOI:** 10.3390/healthcare11152134

**Published:** 2023-07-26

**Authors:** Judy Leung, Kin-Kit Li

**Affiliations:** 1School of Nursing & Health Studies, Hong Kong Metropolitan University, Hong Kong SAR, China; 2Department of Social and Behavioural Sciences, City University of Hong Kong, Hong Kong SAR, China; ben.li@cityu.edu.hk

**Keywords:** spiritual intervention, depression, connectedness, Christian, community

## Abstract

Depression is a common, depleting, and potentially life-threatening disorder. This pilot study examined the feasibility and applicability, reported preliminary evidence for effectiveness, and explored the potential healing mechanisms of a faith-based spiritual intervention for people with depression. The intervention consisted of six weekly sessions focused on restoring a connection with the divine, forgiveness and freedom, suffering and transcendence, hope, gratitude, and relapse prevention. Seven adults with mild or moderate depressive symptoms were recruited. A qualitative evaluation was conducted via focus group discussions, and rating scales were administered at baseline, after the intervention, and at the 3-month follow-up. The mean difference scores of the treatment’s effect over time were analyzed using Friedman’s ANOVA. The themes identified by the focus group included the meaning of the spiritual intervention, the effect of involvement in a spiritual group, and the therapeutic components. The results indicated a significant decrease in the mean scores for depression (PHQ-9) after intervention and at the 3-month follow-up. Participants expressed their improvement in terms of increased knowledge about depression, enhanced coping mechanisms, and improved self-esteem. The preliminary evidence suggested that the faith-based spiritual intervention was effective in reducing depressive symptoms and also helped participants develop a greater sense of connection with themselves, others, and their environment.

## 1. Introduction

Depression is a common and serious mental health issue affecting over 300 million people of all ages around the world [1]. This specific problem is considered a major factor in causing disabilities and is recognized as the primary source of global disability [2].

Individuals experiencing depression often endure a prolonged state of sadness or disinterest in the activities they typically find enjoyable. They may also experience a lack of energy, alterations in their appetite, changes in their sleeping patterns, feelings of anxiety, difficulties in concentration, indecisiveness, restlessness, a sense of worthlessness, guilt, or hopelessness, and even suicidal thoughts. These symptoms may persist for at least two weeks and can lead to impaired social and occupational functioning [3]. Apart from the debilitating effects of the illness that hamper an individual’s quality of life, depression may also lead to self-harm or suicide. It is a major contributor to suicidal deaths, making it one of the top 20 leading causes of death [2].

Despite the availability of various mental health interventions, depression is one of the most under-reported and undertreated mental disorders [4]. This may be due to ignorance or a lack of knowledge about one’s mental health issues. People’s belief in mental health or stigmatization might also hinder help-seeking behaviors [5].

The global burden of mental health disorders continues to be a significant public health concern [1]. Despite the availability of various mental health interventions, many individuals still struggle to access effective treatment. Moreover, the current approaches have not been able to fully address the complex needs of individuals with mental health disorders [6]. Therefore, there is a pressing need to explore novel evidence-based interventions that can be integrated into existing mental health services to improve access to effective treatment and help individuals achieve better mental health outcomes [7,8].

Traditional treatments for depression include pharmacological interventions, psychotherapy, or a combination of both. Both of them are proven to be effective treatments for depression, and the evidence for the treatment efficacy of pharmacological interventions is considered higher than that of psychotherapy [9]. However, some people may experience unpleasant side effects from antidepressants that impair their quality of life, leading to the discontinuation of the treatment [10]. Furthermore, prolonged use of antidepressant treatment is associated with a loss of symptom reducing efficacy and may cause a return of full depressive symptoms [9]. Consequently, an increasing number of patients with depression have explored non-pharmacological alternatives [10]. Examples of possible interventions include psychoeducation, physical exercise, problem-solving therapy, guided self-help and behavioral activation treatment, participative decision-making, cognitive behavior therapy, electroconvulsive therapy, phototherapy, repetitive transcranial magnetic stimulation, supplements [11,12], and complementary and alternative therapies [13]. Religious/spiritual interventions are often considered a form of complementary therapy. Here, we focused on Christianity as the spiritual intervention since it is one of the most influential religions. It is the largest of the world’s religions, and it is estimated that about one-third of the world’s people are Christians [14,15].

Religion and spirituality are topics of increasing interest. The spiritual aspect is part of holistic care. Holistic care recognizes people with mental health problems as whole persons with interrelated psychological, social, physical, and spiritual needs [16]. Mental health professionals have shown an increasing interest in psychosocial and spiritual care for the recovery of mental health in the past few decades [17,18].

Studies examining the relationship between religiosity and improved mental health have demonstrated that individuals who are more religiously engaged tend to experience fewer symptoms of depression and may recover more quickly from depressive disorders than those who are less religiously involved [19,20,21]. Although studies examining religion/spirituality and depression generally suggest positive associations, there are also potential negative effects of spirituality and religiosity on depression. Individuals may feel guilt and shame if they cannot meet the standards of behavior expected by their religion/spiritual community, which may lead to anxiety and depression [22]. Individuals who practice religion but utilize negative religious coping mechanisms, such as blaming God for their hardships, have been linked to increased depressive symptoms [19]. Healthcare professionals need to be sensitive to clients’ spiritual needs, which might impact the progress of their illness.

Previous studies have indicated a significant positive association between religion/spirituality and better mental health. The positive results of the reviewed faith-based interventions also suggested their potential for practical use. Although the benefits of those programs have been noted, the exact effects and mechanisms of healing depression were not demonstrated in those studies.

Ministers have been providing counseling and care to church members all along. The findings of a cross-sectional survey from the National Comorbidity Survey in the United States indicated that a higher percentage of people sought help for their mental disorders from clergy (23.5%) compared with psychiatrists (16.7%) or general medical doctors (16.7%) [23]. According to the Hong Kong Mental Morbidity Survey (HKMMS), the estimated prevalence rate for Common Mental Disorders (CMD) was 13.3%, with mixed anxiety and depressive disorder being the most commonly diagnosed conditions. Among individuals in Hong Kong who were identified as having CMD, only 26% had sought mental health services within the past year, while 10% had consulted general practitioners or family physicians [24]. The situation is suboptimal, and no study has investigated whether individuals with depression seek help from spiritual/religious sources.

Spirituality can be regarded as being connected to everything and knowing that it goes beyond every person to something much bigger than all of us together [25,26]. Spiritual connectedness may be perceived as having a vertical dimension of relationship with the transcendent/God and a horizontal dimension with others, thus resulting in a change in values, beliefs, and interactions with others [27,28].

Empirical studies have consistently found that there is a positive correlation between forgiveness and the alleviation of depressive symptoms [29,30]. Individuals may feel guilty about not fulfilling some responsibilities. The ability to forgive not only the offender but also ourselves can free us from being victims, which is true freedom [31].

Spirituality and religion can help the individual find meaning in their suffering. Finding meaning in suffering is a transcendent experience. The discovery of meaning in the experience of illness helps individuals reconcile their distress and leads to a transcendence of suffering [32].

Hope is inversely associated with depression [33,34,35]. Spirituality and hope can come from patients’ inner strength, which can be enhanced by helping them explore the meaning of life and deep personal values, encouraging them to pursue their life goals, and teaching them skills to regulate their emotions and live healthy and enjoyable lives [36].

Gratitude could be included, as evidence has shown an inverse relationship between gratitude and depression [37,38,39]. As relapse is a significant risk among people treated for depression, specific interventions are needed to address and minimize this risk. The spiritual intervention program was developed to help people with depression reconnect with a higher power, others, and themselves. This will be discussed in the methods section.

Depression is an experience of connections and disconnections in the journey of life [40]. This difficult experience of depression disconnects people from themselves, others, and society. Furthermore, it creates a sense of spiritual disconnection and may result in a lack of personal direction, loneliness, and spiritual crises [41,42]. Many physical and emotional problems can be created by negative states arising from feelings of disconnection from life [43].

Several gaps of knowledge in the literature were noted: (1) the lack of evidence on the healing mechanism of spiritual interventions; (2) the lack of studies using a methodologically rigorous design, e.g., a randomized controlled trial (RCT), to investigate the efficacy of spiritual interventions for persons with depression; and (3) the limited number of studies using spiritual interventions for persons with depression in the Chinese context.

The lack of research-based work in these areas served as the motivation for this study. A spiritual-based program providing participants with information on spiritual coping was developed so that they could practice it in their daily lives. Recovery is a journey of self-discovery, self-renewal, and transformation that involves readjusting one’s attitudes, feelings, perceptions, and beliefs about oneself, others, and life [42]. Spiritual renewal and recovery are closely connected. Thus, the rationale of the program matched the needs of the target population.

### Aims

The primary objective of this study was to examine the feasibility and acceptability of this faith-based spiritual intervention before conducting a full-scale RCT. In addition, we also collected preliminary evidence to assess the effectiveness and explore the potential healing mechanisms of this intervention.

To sum up, the objectives of the pilot study were:To explore the acceptability and feasibility of a spiritual intervention for people with depression;To examine the effects of this intervention on depressive symptoms, hope, meaning in life, self-esteem and social support, anxiety levels, and daily spiritual experience;To explore the participants’ views on the healing mechanisms of the intervention.

## 2. Materials and Methods

### 2.1. Study Design

A pre- and post-test group approach with a 3-month follow-up was used in this study. It was conducted from 17 April to 22 May 2021 in a non-government organization. Quantitative data were collected from the participants on three occasions: baseline, after the intervention, and at a 3-month follow-up session. Qualitative data were collected via a focus group discussion immediately after the intervention. This explanatory sequential mixed-method design provided reflections and ideas for further analysis [44].

### 2.2. Participants

The study enrolled 7 adults (6 females and 1 male) with mild or moderate depressive symptoms. Five of them were referred by an NGO with a Christian faith background, and two of them were recruited through personal networks. The inclusion criteria were Hong Kong Chinese residents who could communicate in Cantonese, were between the ages of 18 and 64 years, had no objection to rituals of the Christian faith, scored between 5 and 14 out of 27 on the Patient Health Questionnaire-9 (PHQ-9), and stated their willingness to comply with the trial’s protocol. The details of the PHQ-9 can be found in the section on measurement of outcomes. The exclusion criteria were those who had received any form of psychotherapy in the past 3 months, those with significant cognitive impairment, a lifetime history of psychosis that would make them unable to understand and follow instructions, those with a strong suicidal risk, or those who had adjusted their medication (antidepressant) within the past 3 months.

The recruitment of participants was facilitated by the organization according to the inclusion criteria. Participants were first screened by telephone to determine whether preliminary inclusion/exclusion criteria were met. Then, they would attend a psychiatric interview with an investigator who was an experienced psychiatric nurse to rule out cognitive impairment or suicidal thoughts, which would be excluded. A psychiatric assessment was performed for every participant prior to the commencement of the pilot study.

### 2.3. Ethical Considerations

Ethical approval to conduct the study was obtained from the university (No. 14-2020-16E) and the participating NGO. Written consent from the participants was obtained prior to the commencement of the study. Participants were briefed about the purpose and procedure of the study, and their right to withdraw. The confidentiality of the participants was ensured, since no identifiable information was collected in the questionnaires. The data obtained from the participants was stored in a locked filing cabinet, and any data stored on a USB stick was password-protected.

### 2.4. Intervention

The intervention was based on a modified theoretical framework that combined spirituality/religiosity in a group therapy approach, a cognitive behavioral therapy framework, and a positive psychology intervention (Figure 1).

Cognitive reframing and behavioral activation were spiritually integrated into the intervention. These are the core skills of cognitive behavioral therapy [45]. This approach focuses on rationality and the avoidance of negative and irrational thinking that affects one’s emotions and behaviors. Scriptures were recited meditatively or prayerfully (behavior) and also used as a way of changing one’s perspective (cognition).

The strategies and skills of the positive psychology intervention helped individuals look at life with optimism, appreciate the present, accept and make peace with the past, be more grateful and forgiving, and look beyond the momentary pleasures and pains of life. There is evidence that a positive psychology intervention approach can cultivate positive feelings, behaviors, or cognitions, enhance well-being, and ameliorate depressive symptoms [46].

Participants can experience the therapeutic elements of altruism, universality, instillation of hope, cohesiveness, catharsis, etc. when an intervention is delivered in a group setting [47]. Peer support from the group can also enhance the participants’ recovery journey. The Christian approach to the spiritual intervention included utilizing various elements of the faith, such as reading Bible verses, prayer, singing hymns, and mutual support within the group, as part of the framework.

The intervention program took place once a week for 2 h over six weeks (Table 1). Each session had a similar format and included three parts. The first part was around 15 min for singing hymns, a review of the homework, and sharing the participants’ application of the topic learned in the past week. The second part was around 90 min with a 10 min break. It focused on activities and a discussion on the theme of that week. The last part (15 min) was a wrap-up of the session and the introduction of the homework. Each session ended with a prayer to instill hope and a spiritual connection.

The first and last sessions focused on introducing and terminating the group intervention. The second to fifth sessions were organized in a structured format in which specific themes were presented with sufficient time for processing, discussion, and practice.

In order to ensure the efficacy, clinical appropriateness, and feasibility of implementing the program, the protocol was sent to a panel of experts, including a psychiatrist, psychiatric nurses, social workers, and academic scholars, for their comments. On the basis of the feedback from these experts, the protocol was modified and revised, and its validity was strengthened.

### 2.5. Measurement

Quantitative data were collected at three time points: at baseline (T0), after the intervention on Week 6 (T1), and at the 3-month follow-up on Week 18 (T2). Qualitative data were collected via a focus group discussion.

#### 2.5.1. Acceptability and Feasibility

A narrative description was adopted to evaluate the study’s feasibility, including aspects of recruitment and retention, protocol adherence, and stakeholder acceptability. Recruitment and retention were evaluated using recruitment logs. A qualitative approach was used to determine adherence to the protocol, the acceptability of the recruitment process, assessments, treatment delivery, and any difficulties encountered.

#### 2.5.2. Measurement of the Outcomes 

The indicators for the effectiveness of this intervention were assessed using a questionnaire instrument. The questionnaire consisted of eight parts, with the measures described below.

Part A: Demographic data. This measured the personal details of the participants and included questions on gender, age, religious affiliation/spirituality, educational status, employment status, and past history of psychiatric treatment.

Part B: Spirituality. The Daily Spiritual Experience Scale (DSES) is a validated self-reported questionnaire consisting of 16 items that assess an individual’s everyday experiences of transcendence [48]. The first 15 items are rated on a Likert scale from 1 (never or almost never) to 6 (many times a day), while the final item has four possible responses ranging from 1 (not close) to 4 (as close as possible). The scores are totaled, with higher scores indicating greater levels of spirituality, ranging from 16 to 94 [49]. The DSES has demonstrated satisfactory reliability and validity and has been translated into over 40 languages, including a Chinese version with a reported Cronbach’s alpha of 0.97 [50,51].

Part C: Hope. The State Hope Scale (SHS) is a self-reported instrument consisting of six items used to measure an individual’s level of hope regarding ongoing events in their lives [52]. It is an 8-point Likert scale ranging from 1 (definitely false) to 8 (definitely true), with odd-numbered items measuring pathway thinking (planning ways to achieve goals) and even-numbered items measuring agency thinking (goal-directed determination). Scores can range from 6 to 48, with higher scores indicating greater levels of hope. The SHS has been reported to be a reliable and valid measure of hope, with factor analyses confirming the two factors of agency and pathways [52]. The Chinese version of the scale has shown high internal consistency [53,54].

Part D: Meaning in life. The Meaning in Life Questionnaire (MLQ) is a 10-item questionnaire that measures two subscales: the presence of meaning in life and the search for meaning in life [55]. It is a 7-point Likert scale ranging from 1 (absolutely untrue) to 7 (absolutely true), with five items for each subscale. The items are summed, yielding a range from 5 to 35 for each subscale, with higher scores indicating a strong presence of or search for meaning in one’s life. The MLQ has been translated into different languages, and the Chinese version of the questionnaire has been found to have the same factor structure as the original version among caregivers in Hong Kong [56].

Part E: Self-esteem. The Rosenberg Self-Esteem Scale (RSES) is a widely used self-reported instrument consisting of 10 items that measure an individual’s level of self-esteem [57]. It is a 4-point Likert scale ranging from 1 (strongly disagree) to 4 (strongly agree), with five negatively worded items being reverse-scored. The total score ranges from 10 to 40, with higher scores indicating greater levels of self-esteem. The reliability and validity of the Chinese version of the RSES have been found to be satisfactory [58,59].

Part F: Social support. The Multidimensional Scale of Perceived Social Support (MSPSS) is a widely used 12-item self-administered measure of social support [60]. It is a 7-point Likert scale ranging from 1 (very strongly disagree) to 7 (very strongly agree), with the mean score of the total scale calculated by summing the scores across all 12 items and dividing the result by 12. The mean score ranges from 1 (lowest) to 7 (highest). The Chinese version of the MSPSS has demonstrated internal consistency and reliability, with composite reliability values greater than 0.7 [61].

Part G: Depression. The Patient Health Questionnaire 9 (PHQ-9) is a self-reported questionnaire designed to assess the nine symptom-based criteria for diagnosing depressive disorder according to the DSM-IV [62]. The scores range from 0 to 27, with scores of 0–4 indicating no or minimal depressive symptoms, 5–9 indicating mild depressive symptoms, 10–14 indicating moderate depressive symptoms, and 15–27 indicating severe depressive symptoms. The PHQ-9 has been found to have good sensitivity and specificity, with Cronbach’s alpha reported to be 0.86–0.89 [63].

Part H: Anxiety. The General Anxiety Disorder (GAD-7) is a 7-item self-administered questionnaire used to screen the severity of general anxiety disorder, with scores ranging from 0 to 21 [64]. Scores of 0–5 indicate mild anxiety, 6–10 indicate moderate anxiety, 11–15 indicate moderately severe anxiety, and 16–21 indicate severe anxiety. The Chinese version of the GAD-7 has been found to be a reliable and efficient instrument [65].

#### 2.5.3. Focus Group

All of the participants were invited to a focus group after the intervention on Week 6 (T1) in order to understand the perceived strengths, limitations, and difficulties of the spiritual intervention program. Focus groups can be used to explain the results found through other data collection methods, and are especially helpful for explaining findings that appear to be counterintuitive or conflicting [66]. A semi-structured interview guide was developed according to the study’s objectives. It contained six guiding open-ended questions to explore the participants’ perspectives on the role of spirituality in depression (Appendix A). The focus group interview was videotaped with the consent of the participants.

### 2.6. Treatment Fidelity

The spiritual intervention was based on a theoretical framework of spirituality/religiosity and the symptoms of depression. Standardization in the delivery of the spiritual intervention was enabled by producing a written protocol for the spiritual intervention. All the sessions were videotaped, and a checklist (Appendix B) was developed for self-monitoring of the intervention’s sessions [67,68].

### 2.7. Data Analysis

Quantitative data were recorded, managed, and analyzed using the IBM Statistical Package for Social Science (SPSS) v 26. An intention-to-treat (ITT) principle was used for analyzing the data, and the last observation carried forward (LOCF) method was used to replace missing data [69]. Descriptive statistics for basic demographics, the number of sessions attended by the participants, and details of the intervention were computed. A non-parametric test, Friedman’s ANOVA, was used to analyze the mean difference scores of the treatment’s effects over time, i.e., a comparison of the baseline with the post-intervention scores and those at three-month follow-up.

The focus group Interview was videotaped, transcribed verbatim, and analyzed using content analysis. A systematic approach known as content analysis was utilized to explore large quantities of textual information in an unobtrusive manner. This involved coding and categorizing the information to identify trends and patterns in the use of words, their frequency, and relationships, as well as the structures and discourses of communication [70].

The first author of this article immersed herself in the data by reading and re-reading the transcripts. The data were analyzed by the author and validated by an independent person who is a PhD student with experience as a research assistant to ensure their rigor and trustworthiness. Trustworthiness is essential in each phase of a qualitative data analysis [71,72]. The researcher and the second coder reviewed the analytical process to discuss the coding categories. Any discrepancies were discussed and adjusted to reach a consensus. After comparing all the results of coding, relevant themes and conceptual categories about the meaning of the spiritual intervention, the effect of the spiritual group, therapeutic components, etc. were formed to describe the strengths and limitations of the program.

## 3. Results

### 3.1. Characteristics of the Participants 

Twenty-six adults were being screened for this pilot study. Nine of them were eligible to join. One of them withdrew before the commencement of the study due to health reasons, and another one failed to provide baseline data. Seven adults with mild/moderate depression were recruited, but only six completed the program, and five attended the focus group interview. All of them were middle-aged people. Six out of seven (85.7%) were female. Six of them had been diagnosed with depression, and the other one had a case of bipolar disorder (Table 2). Five of them had full attendance (71.4%), i.e., they attended all the sessions. One of them attended four out of six sessions (14.3%), and one of them dropped out. All of them had received treatment before, which might have been antidepressants and/or psychotherapy. Five of them were still taking antidepressants at the time of the study, but their medication had not been adjusted within the past 3 months or throughout the study. 

The video recordings were reviewed to ensure compliance with the protocol. The score of the checklist for self-monitoring ranged from 79% to 100%, with a mean of 92%. These scores revealed a very satisfactory level of adherence, according to the recommendations of the NIH Behavior Change Consortium [67].

### 3.2. Quantitative Analysis

Quantitative data, including the Patient Health Questionnaire-9 (PHQ-9) and General Anxiety Disorder Questionnaire (GAD-7), were compared at three time points (before and after the intervention and at 3-month follow-up). Since this was an exploratory pilot study with a small sample for planning a future RCT, a two-tailed 0.10 level of significance (α) was used. There was a significant decrease in the level of depression among the participants after the pilot study and at the 3-month follow-up. The results are shown in Table 3. The baseline score was 7.43 (3.409), the post-intervention score was 5.14 (3.532), and the score at 3-month follow-up was 4.86 (4.598). The effect sizes were moderate (Cohen’s *d*: 0.66–0.63, *p*-value = 0.094). For the General Anxiety Disorder Questionnaire (GAD-7), the baseline score was 5.71 (4.751), the post-intervention score was 4.14 (3.436), and the score at 3-month follow-up was 3.43 (3.867). The effect sizes were non-significant, although the effect sizes ranged from small to moderate (Cohen’s *d*: 0.39–0.53, *p*-value = 0.264). A decreasing trend in the mean scores of PHQ-9 and GAD-7 was noted (Figure 2).

The results of other assessment outcomes, including the Daily Spiritual Experience Scale (DSES), State Hope Scale (SHS), Meaning in Life (MLQ), Rosenberg’s Self-Esteem Scale (RSES), and the Multidimensional Scale of Perceived Social Support (MSPSS), were relatively insignificant (Table 3).

### 3.3. Qualitative Analysis

The content analysis of the focus group discussion provided rich data on the participants’ views on the spiritual intervention. This was also reflected by the frequency of words they used during the focus group (Table 4).

The themes identified were the meaning of the spiritual intervention, the effect of involvement in the spiritual group, therapeutic factors/components of the spiritual intervention, and participants’ views on and suggestions for the program. The themes and their related categories and subcategories are shown in Table 5.

#### 3.3.1. Meaning of the Spiritual Intervention

Participants expressed that the spiritual intervention was related to their faith. “Spirituality is related to our faith… faith can play a role in some of the problems we are facing (C2)”. They identified different spiritual dimensions, including the almighty power of God, spiritual guidance, spiritual inspiration, the words of God, spiritual comfort, spiritual experience, and prayer. They also conceptualized the spiritual intervention as a process of spiritual transformation involving spiritual growth, centering, and compassion.

#### 3.3.2. Effects of Involvement in the Spiritual Group 

The experience of participating in the group was highly valued by the participants, who felt accepted and supported by their peers. They described the group as a safe space where they could share their experiences and feelings without fear of judgment. One of the participants said, “Here is just accepted” (A7), while another noted that the group was “the peers among peers” (A27). This sense of belonging and peer support were a significant factor in their positive experience. 

In addition to feeling accepted, the participants also reported a significant increase in their knowledge and understanding of coping with depression and preventing relapse. They felt that the group provided them with practical tools and strategies for managing their symptoms and avoiding negative patterns of thinking and behavior. As one participant noted, “This is exactly the benefit of joining the group…depression—recognizing, dealing with and preventing it…” (C19), while making a gesture to indicate the importance of this learning experience.

The participants also described applying what they had learned in the group to their daily lives. They reported behavioral changes, such as learning how to look at their own problems … and to continuing to apply the skills they had learned …” (B4). They expressed increased coping ability and improvement in their symptoms. This suggests that the group was effective in helping them make meaningful changes in their lives. 

Participants also reported growth in their spiritual lives. They described using a “Spiritual diary… Praying… devotional… asking God” (D21). This aspect of the group was significant for those who felt that spirituality played an important role in their mental health and well-being.

#### 3.3.3. Therapeutic Factors/Components of the Spiritual Intervention

The participants also reported a strong belief in the power of God and his omnipotence: “…first you have to believe that this God is strong…” (C13). They believed in the supernatural power of God: “He knows all things and all things are under His control… God is sitting as King, in charge of all things…” (C38). They described their relationship with God as a “master–servant relationship” (C38) and that of children: “…God calls us God’s people, or children…” (B9). They also firmly believed God was listening to their prayers: “God really listens to prayers” (D19).

The spiritual practices that they considered therapeutic were the use of scriptures, prayer, sharing, and testimonies. They also benefited from the therapeutic factors of belonging to a group with similar backgrounds, sharing, and mutual support. Spiritual advancement through forgiveness, obedience, finding meaning in suffering, and experiencing God was one of the therapeutic factors: “God must have had an intention in allowing me to have this experience” (C25). Finding inspiration through self-understanding and increased insight also contributed to their healing.

#### 3.3.4. Participants’ Views on and Suggestions for the Program

Participants provided feedback on both the logistics and content of the program. With regard to the logistics, they suggest the technical issues need to be addressed. Regarding the content of the program, the participants expressed appreciation for what they had learned. They reported that the program had helped them to “look at their own problems “(B4) and had provided them with practical tools and strategies for coping with depression (C19) and “preventing relapse” (C22). However, they expressed that the duration was a bit short: “Duration of the program a bit short” (A68, A70. A72). They also suggested that a reunion be held.

## 4. Discussion

The primary objective of this pilot study is to assess the feasibility and acceptability of a faith-based spiritual intervention before proceeding to the main study [73]. In addition, we also reported the preliminary evidence for the effectiveness and explored the perceived healing mechanisms of this intervention. These objectives were well met. 

This pilot study showed promising evidence of feasibility and acceptability. Out of 7 participants, 5 attended all sessions, indicating a satisfactory completion rate. Additionally, the program received positive feedback from participants during focus group discussions, and therapeutic factors were identified. 

An Alpha at the 0.1 level was adopted in this small pilot study, and a significant effect was found for the depressive symptoms but not the other measurements. The decreasing trend in the mean score of PHQ-9 and GAD-7 from baseline to after the intervention and for the follow-up at 3 months also indicated the effect of the spiritual intervention on decreasing the depressive symptoms and anxiety levels.

One of the objectives of this pilot study was to examine the preliminary effects of the spiritual intervention protocol. The intervention was considered effective, as perceived by the participants. However, they indicated that the program was a bit short and suggested having a few more sessions. Thus, the content of the program in the main study should be the same, but the duration should be lengthened. This feedback suggests that, while the program was helpful, some participants may have wanted more time to explore the materials and practice the skills they had learned. 

Another point arising from the recruitment was that all the participants were Christians. Although the NGO is a faith-based organization, they also accept non-Christians as members. It is possible that Christians are looking for religious and spiritual healing, as they consider Jesus to be the greatest physician: “A great crowd of people followed him because they saw the signs he had performed by healing the sick” (John 6:2). This was also reflected in the frequency of words that the participants used during the focus group. The most frequently appearing words were God, prayer/pray, Bible, and scripture.

The qualitative evaluation by the focus group revealed that spiritual intervention was conceptualized by the participants as a process of spiritual transformation with different spiritual dimensions, which included almighty power, spiritual guidance, spiritual inspiration, God’s word, spiritual comfort, spiritual advance, spiritual experience, and prayer. They expressed their improvement in terms of knowledge of depression, enhanced coping mechanisms, and self-esteem.

The participants expressed finding spiritual comfort in the group, where they had a chance to vent their feelings. They felt accepted and identified with others who were “in the same boat” during the focus group discussion. The group dynamics helped them to look at their own problems, consider how to share or provide support, and continue learning and applying what they had learned in their daily lives. These findings are consistent with a preliminary qualitative study on the clients and facilitators of a self-help group, which also identified trust, acceptance, mutual support, and encouragement as key factors in the group dynamic [74]. 

The participants had started to connect to others and their environment. Participants’ connection to their higher power was demonstrated by their expressions of having a better relationship with God. As they gained inspiration through self-understanding and increased insight, they started to reconnect with themselves.

The participants created a WhatsApp group after the completion of the intervention. They continue to support each other, and share their difficulties, and some of them have regular prayers together. Through the intervention, the participants were able to reconstruct and enhance their sense of connectedness to various aspects of life, including themselves, others, their environment, their way of life, and a larger meaning and purpose.

The results of the qualitative analysis indicate participants’ overall satisfaction with the treatment delivery, and the study was conducted according to the planned protocol. These findings suggest that this faith-based spiritual intervention program is feasible and acceptable for the target population in the community. 

### 4.1. Limitations

The current study was limited by its small sample due to the difficulty in recruiting participants from a single NGO, which decreased the generalizability of the findings. Moreover, all the participants had a belief in Christianity, so their perceptions and responses to spirituality may be different from people with a non-Christian background. Apart from that, all the participants were middle-aged people, and their preferences may also be different from those of a younger population. Furthermore, biological indicators such as blood or saliva were not being used due to the practical challenges and potential invasiveness of collecting blood or saliva samples in this population. The results must be interpreted with caution. Another limitation of this study is the lack of a control group, so we cannot be sure that the significant differences observed were due to placebo effects.

### 4.2. Implications and Suggestions for the Main Study

Based on the results of the pilot study, modifications will be made to the main study. The recruitment of participants will be expanded to include a diversity of sources, such as different NGOs, local churches, and tertiary institutions, to enhance the generalizability of the research. A robust RCT will be conducted for the main study. Future studies can consider using biological indicators such as blood or saliva markers in the appropriate populations, as it would be more objective. Furthermore, clinical trials comparing spiritual intervention with other psychotherapeutic approaches can also be explored in the long run.

The healing mechanism could be further explored and validated by a qualitative evaluation from more focus group sessions in the main study. The program’s duration will be extended from six sessions to eight sessions to allow more time for the participants to share and interact. An online intervention with the same content can be adopted in cases where a face-to-face intervention is not possible (e.g., during a pandemic). Additionally, a reunion will be held after an interval of 3 months during the follow-up period.

## 5. Conclusions

This study adopted an explanatory sequential mixed-methods research design by including quantitative and qualitative measures of the outcomes. The findings suggested that this faith-based spiritual intervention program is feasible and acceptable for the target population in the community, given the high completion rate and attendance rate and the positive feedback collected from the focus group.

The collection of qualitative data from the clients’ perspective was an asset of this pilot study, providing valuable information on the spiritual intervention from the participants’ perspectives. The therapeutic factors of the spiritual intervention identified by the qualitative data indicated the healing mechanism of connectedness. The study provides preliminary support for the effectiveness of the intervention based on both quantitative and qualitative data.

## Figures and Tables

**Figure 1 healthcare-11-02134-f001:**
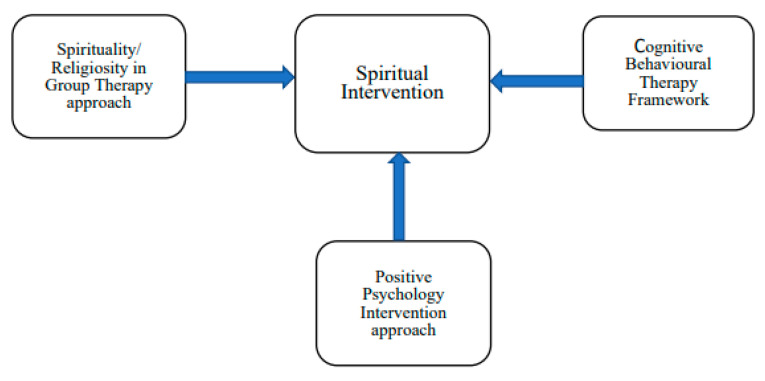
Development of a spiritually based intervention program.

**Figure 2 healthcare-11-02134-f002:**
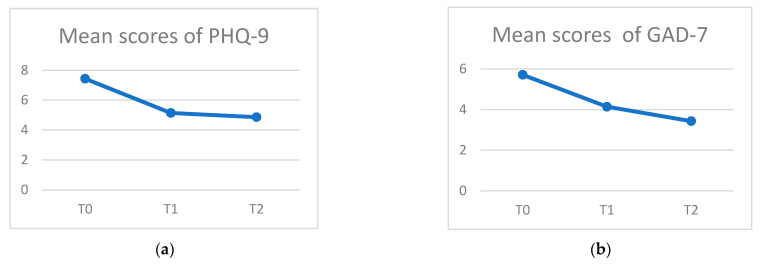
Mean scores of PHQ-9 and GAD-7 over time: (**a**) mean scores of PHQ-9 at T0, T1, and T2; (**b**) mean scores of GAD-7 at T0, T1, and T2.

**Table 1 healthcare-11-02134-t001:** Content of the spiritual intervention program.

Session	Content
1	Spirituality, mental health, and depression
2	Connectedness
3	Forgiveness and freedom
4	Suffering and transcendence
5	Hope and gratitude
6	Relapse prevention, review of the materials, and celebration

Weekly assignments, assignments, daily meditation, and prayer were included in the program.

**Table 2 healthcare-11-02134-t002:** Demographic characteristics of the participants.

Characteristics	Number	%
Age		
46–55	3	42.9
56–64	4	57.1
Gender		
Male	1	14.3
Female	6	85.7
Marital status		
Single	3	42.9
Married	3	42.9
Divorced	1	14.3
Religion		
Protestant Christianity	7	100
Educational background		
Secondary	6	85.7
Doctoral	1	14.3
Occupational status		
Unemployed	1	14.3
Retired	4	57.1
Other	2	28.6
Received treatment		
Yes	7	100

**Table 3 healthcare-11-02134-t003:** Comparison of outcome scores at baseline, post-intervention, and 3-month follow-up.

Assessment Outcome	Pre-Intervention (T0) Mean (SD)	Post-Intervention (T1) Mean (SD)	3-Month Follow-Up (T2) Mean (SD)	*p*-Value	Cohen’s *d* T0 vs. T1	Cohen’s *d* T0 vs. T2
PHQ-9	7.43 (3.409)	5.14 (3.532)	4.86 (4.598)	0.094 *	−0.66	−0.63
GAD-7	5.71 (4.751)	4.14 (3.436)	3.43 (3.867)	0.264	−0.39	−0.53
DSES	62.86 (21.075)	65.57 (21.678)	63.71 (20.131)	0.580	0.13	0.04
SHS	30.29 (9.196)	26.71 (11.309)	29.43 (9.744)	0.834	−0.34	−0.09
MLQ-Presence	26.14 (5.398)	20.57 (9.071)	22.86 (6.594)	0.170	−0.75	−0.54
MLQ-Search	25.57 (10.277)	20.29 (8.499)	21.00 (7.937)	0.568	−0.56	−0.49
RSES	30.00 (3.512)	31.86 (5.956)	32.14 (5.242)	0.108	0.38	0.48
MSPSS	4.89 (1.103)	4.42 (0.715)	4.39 (0.938)	0.607	−0.51	−0.49

Note: PHQ-9, Patient Health Questionnaire-9; GAD-7, General Anxiety Disorder Questionnaire-7; DSES, Daily Spiritual Experience Scale; SHS, State Hope Scale; MLQ-Presence, the Meaning in Life Questionnaire — presence of meaning in life; MLQ-Search, the Meaning in Life Questionnaire—search for meaning; RSES, Rosenberg’s Self-Esteem Scale; MSPSS, Multidimensional Scale of Perceived Social Support. * *p* < 0.1. T0 vs. T1 = values at the baseline versus the values after the intervention. T0 vs. T2 = values at the baseline versus the values at the 3-month follow-up. Effect size (Cohen’s d): small = 0.20; moderate = 0.50; large = 0.80.

**Table 4 healthcare-11-02134-t004:** Frequency of words used in the focus group.

Words	Frequency
God	86
Prayer/pray	31
Bible	18
Support	17
Scripture	17
Sharing/share	14
Faith	10

**Table 5 healthcare-11-02134-t005:** Summary of the themes, categories, and subcategories.

Theme 1: Meaning of the Spiritual Intervention	
Category 1: Faith	
Category 2: Spiritual dimension	Subcategory 1: God’s almighty power
	Subcategory 2: Spiritual guidance
	Subcategory 3: Spiritual inspiration
	Subcategory 4: Words of God
	Subcategory 5: Spiritual comfort
	Subcategory 6: Spiritual experience
	Subcategory 7: Prayer
Category 2: Spiritual transformation	Subcategory 1: Spiritual growth
	Subcategory 2: Centering
	Subcategory 3: Compassion
Theme 2: Effect of the spiritual group	
Category 1: Group experience	Subcategory 1: Acceptance
	Subcategory 2: Direction and focus
	Subcategory 3: Peer support
	Subcategory 4: Genuine sharing
	Subcategory 5: Better relationship with God
Category 2: Increased understanding	Subcategory 1: Knowledge
	Subcategory 2: Coping with depression
	Subcategory 3: Relapse prevention
	Subcategory 2: Gratitude
	Subcategory 3: Preparation in advance
Category 4: Spiritual life	Subcategory 1: Communication with God
	Subcategory 2: Devotion
Category 5: Improvement of symptoms	Subcategory 1: Improvement in general condition
	Subcategory 2: Coping
	Subcategory 3: Increased self-esteem
Theme 3: Therapeutic factors/components of the spiritual intervention	
Category 1: Belief	Subcategory 1: Power of God
	Subcategory 2: God’s omnipotence
	Subcategory 3: Relationship with God
	Subcategory 4: God is listening
Category 2: Spiritual practice	Subcategory 1: Use of scripture
	Subcategory 2: Prayer
	Subcategory 3: Sharing
	Subcategory 4: Testimonies
Category 3: Group factors	Subcategory 1: Similar background
	Subcategory 2: Emotional catharsis
	Subcategory 3: Support
Category 4: Spiritual advancement	Subcategory 1: Forgiveness
	Subcategory 2: Obedience
	Subcategory 3: Meaning in suffering
	Subcategory 4: Experiencing God
Category 5: Inspiration	Subcategory 1: Self-understanding
	Subcategory 2: Increased insight
Theme 4: Participants’ views on and suggestions for the programme	
Category 1: Suggestions for the programme	Subcategory 1: Logistics of the program
	Subcategory 2: Program’s content and duration
	Subcategory 3: Future suggestion

## Data Availability

The data of this study contain information that would compromise the privacy of the research participants and are not publicly available.

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
