# Peer review of "Faith-Based Spiritual Intervention for Persons with Depression: Preliminary Evidence from a Pilot Study"

_healthcare, 2023, doi:10.3390/healthcare11152134_

Round 1

Reviewer 1 Report

Thank you very much for giving me the opportunity to review this interesting manuscript. In my opinion, the manuscript is well written and the methodology used is correct. They performed a spiritual intervention to see the results in patients with depression. The authors make it clear that it is a pilot study. My criticism of the study is the small number of individuals included. However, this pilot study can be used for future studies with a larger number of subjects included.

Author Response

Please find my reply in the attached file.

Thanks.

Reviewer 2 Report

In the presented Article, Judy Leung and K.-K. Li. describe the pilot study, performed on 7 adult depressive patients, suggesting the beneficial effect of spiritual, and religious meetings on depressive symptoms measured with PHQ-9 and GAD-7 scales. The scientific approach of the Article is interesting. However, due to not clear aim of the study, methodology descriptions and result discussion in the context of future clinical trials based on the received pilot study, I cannot recommend this Article for publication in its current shape.

Main comments:

1) The aim # 1 of the study presented on page 3 of the manuscript is not in agreement with the pilot study definition (“A small-scale test of the methods and procedures to be used on a larger scale”), because authors plan to check whether applied treatment is effective instead of to assess feasibility/acceptability of a new approach to be used in a larger scale study. Moreover aim #2 is not clear and readers cannot find which of the presented results, and in which way show the participants’ perspectives on spiritual interventions. Please adjust the aims to the data presented in the manuscript and pilot study requirements. Additionally, please improve the aims description and their consistency throughout the manuscript.

2) There is no clear statement in the Article, about which kind of clinical studies will be performed based on obtained pilot studies. Please complete.

3) Effectiveness of spiritual meetings, the authors describe in the manuscript seems to be similar to, clinically useful, group psychotherapy. Can you confront the antidepressive effectiveness of your method with psychotherapeutic approaches used in clinical practice?

4) Characteristics of the study’s participants is not clear. Table 2 says that their age is 46-64 years while in subsection 2.2 (page 4) the reader can find 18-64. Please correct.

5) Characteristics of the study’s participants do not specify clearly whether the studied patients were using antidepressants/which kind of antidepressant or not. Please complete.

6) the title of the manuscript does not reflect the content of the article. Please adjust the title to presented and discussed data. If necessary, tone down the title of the manuscript, and correct data descriptions and interpretation.

7) The full name of the last author should be used instead of abbreviation "K.-K." Please improve.

Minor editing of English language is required. I recommend authors to ask a colleague who is a native English speaker to review your manuscript for clarity.

Author Response

Please see my reply in the attached file.

Thanks.

Reviewer 3 Report

The current pilot research was focused on the development of a Spiritual Intervention Program for Persons with Depression,

The employment of non-pharmacological strategies is of paramount importance in the management of depression and anxiety.

Below are some comments to the authors

1.     The study recruited only 7 participants, I have doubts regarding the validity of the study since the scores of depression and anxiety are entirely subjective, are the any supportive studies for this sample size? Is there any possibility to increase the sample size?

2.     The study did not take any blood or saliva markers, for example salivary cortisol is a marker of stress that is a leading cause for depression and anxiety

3.     The diagnosis of the patients was self-assessed, although it can be acceptable for cohort studies, however the lack of medical or psychiatric assessment of the patients remains a big question

4.     The Chistian faith is a deep -rooted faith founded by Jesus Christ, “the word of the living God”. In Table 4, we can not find the word Jesus Christ, was it used interchangeably with the word “GOD”?

5.     The deep-rooted Christian faith goes back to almost 2000 years, where the sacraments were founded by the Lord Jesus Christ himself namely the sacrament of reconciliation and the sacrament of Eucharist (please refer to the Bible: Gospel of John chapter 20 and Chapter 6), why these two major interventions were excluded?

Minor English language editing may be required

Author Response

(The authors gave the same response as above.)

Round 2

Reviewer 2 Report

The authors addressed all my comments and improved the manuscript based on them. I have no additional comments.

Reviewer 3 Report

Thank you for your work